# Vision-Enhanced Semantic Entity Recognition in Document Images via Visually-Asymmetric Consistency Learning

**Hao Wang[1], Xiahua Chen[1], Rui Wang[2]\* and  Chenhui Chu[3]**

[1]School of Computer Engineering and Science, Shanghai University, China
[2]Department of Computer Science and Engineering, Shanghai Jiao Tong University, China
[3]Graduate School of Informatics, Kyoto University, Japan
wang-hao@shu.edu.cn, cxh110029@gmail.com
wangrui12@sjtu.edu.cn, chu@i.kyoto-u.ac.jp

## Abstract

Extracting meaningful entities belonging to pre-defined categories from Visually-rich Form-like Documents (VFDs) is a challenging task. Visual and layout features such as font, background, color, and bounding box location and size provide important cues for identifying entities of the same type. However, existing models commonly train a visual encoder with weak cross-modal supervision signals, resulting in a limited capacity to capture these non-textual features and suboptimal performance. In this paper, we propose a novel **V**isually-**A**symmetric co**N**sisten**C**y **L**earning (VANCL) approach that addresses the above limitation by enhancing the model's ability to capture fine-grained visual and layout features through the incorporation of color priors. Experimental results on benchmark datasets show that our approach substantially outperforms the strong LayoutLM series baseline, demonstrating the effectiveness of our approach. Additionally, we investigate the effects of different color schemes on our approach, providing insights for optimizing model performance. We believe our work will inspire future research on multimodal information extraction.

## 1 Introduction

Information extraction (IE) for visually-rich form-like documents (VFDs) aims to handle various types of business documents, such as invoices, receipts, and forms, that may be scanned or digitally generated. This task has attracted significant attention from the research and industrial communities (Xu et al., 2020; Garncarek et al., 2021; Gu et al., 2021; Li et al., 2021a,b). As shown in Figure 1, the goal of IE for VFDs is to identify and extract meaningful semantic entities, such as company/person names, dates/times, and contact information, from serialized OCR (Optical Character Recognition) output in the documents. Since a single modality may not capture all the semantic information

---
\*Corresponding author

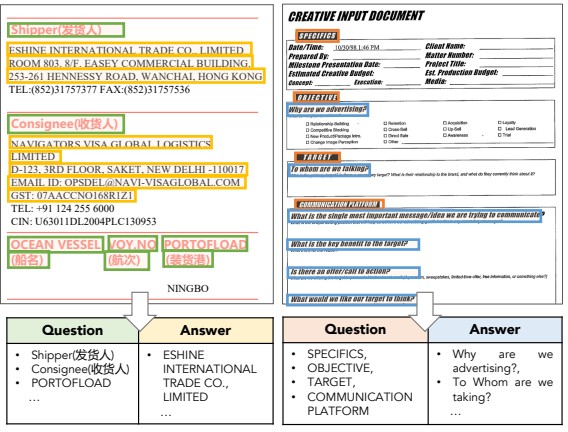

Figure 1: Motivation of this work. Semantic entities of the same type often have similar visual and layout properties, such as the same or similar font, background, color, and the location and size of bounding boxes, providing important indications for recognizing entities and their types. Despite the importance of these properties, existing LMMs for information extraction in VFDs often rely on a limited visual encoder that cannot fully capture such fine-grained features. Therefore, this work focuses on incorporating these visual priors using colors into the task of IE for VFDs.

present in the document, it is necessary to leverage multimodal information, including text, spatial, and visual data. Therefore, large-scale pretrained Multimodal Models (LMMs) (Gu et al., 2021; Li et al., 2021a; Appalaraju et al., 2021; Xu et al., 2021; Huang et al., 2022; Wang et al., 2022a; Lee et al., 2022), which are models that can process multiple modalities of data, have emerged as the dominant approach in IE for VFDs in recent years. State-of-the-art LMMs integrate advanced computer vision models (Ren et al., 2015; He et al., 2016) within BERT-like architectures (Devlin et al., 2019) to leverage spatial and visual information along with text and learn multimodal fused representations for form-like documents. However, these representations are biased toward textual and spatial modalities (Cooney et al., 2023) and have limited per-

formance, especially when the data contains richer visual information. This is because the visual encoder in these models usually plays a secondary role compared to advanced text encoders.

There are two problems with the visual encoder in previous LMMs. First, these models only impose coarse-grained cross-modal constraints during pre-training (e.g., text-image, word-patch, and layout-text alignment (Xu et al., 2021; Huang et al., 2022; Wang et al., 2022a)) to enhance feature extraction from the visual channel, but this does not capture sufficient fine-grained visual features, leading to limited performance and underutilization of prior knowledge in vision. Second, the visual encoders in previous LMMs have weaker representational capabilities than those in the latest Optical Character Recognition (OCR) engines because they do not consider the intermediate tasks such as text segment detection and bounding box regression, which are important for accurately localizing and extracting fine-grained visual features.

To address the issues mentioned above, inspired by recent works on consistency learning (Zhang et al., 2018; Miyato et al., 2019; Xie et al., 2020; Lowell et al., 2021a; Liang et al., 2021; Chen et al., 2021b), we propose a novel vision-enhanced training approach, called **V**isually-**A**symmetric co**N**sisten**C**y **L**earning (VANCL). By incorporating color priors with category-wise colors as additional cues to capture visual and layout features, VANCL can enhance the learning of unbiased multimodal representations in LMMs. Our approach aims to jointly consider the training objectives of the text segment (or bounding box) detection task and the entity type prediction objective, thereby bridging the preprocessing OCR stage with the downstream information extraction stage.

VANCL involves a standard learning flow and an extra vision-enhanced flow. The inputs are asymmetric in the visual modality, with the original document images input to the standard flow, while images to the vision-enhanced flow are the synthetic painted images. The vision-enhanced flow also can be detached when deploying the model. During training, we encourage the inner visual encoder to be as strong as the outer visual encoder via consistency learning. As a result, VANCL outperforms existing methods while (1) extremely little manual effort to create synthetic painted images, (2) no need to train from scratch, whilst (3) no increase in the deployment model size.

Our contributions can be summarized as follows:

- We propose a novel consistency learning approach using visually-asymmetric inputs, called VANCL, which effectively incorporates prior visual knowledge into multimodal representation learning.

- We demonstrate the effectiveness of VANCL by applying it to the task of form document information extraction using different LMM backbones. Experimental results show that the improvements using VANCL are substantial and independent of the backbone used.

- We investigate how different color schemes affect performance, and the findings are consistent with cognitive psychology theory.

## 2 Problem Formulation

We treat the Semantic Entity Recognition (SER) task as a sequential tagging problem. Given a document $\mathcal{D}$ consisting of a scanned image $\mathcal{I}$ and a list of text segments within OCR bounding boxes $\mathcal{B} = \{b_1, \ldots, b_N\}$, we formulate the problem as finding an extraction function $F_{\text{IE}}(\mathcal{D} : \langle \mathcal{B}, \mathcal{I} \rangle) \rightarrow \mathcal{E}$ that predicts the corresponding entity types for each token in $\mathcal{D}$. The predicted sequence of labels $\mathcal{E}$ is obtained using the "BIO" scheme – {*Begin, Inside, Other*} and a pre-defined label set. We train our sequential tagging model based on pretrained LMMs and perform Viterbi decoding during testing to predict the token labels. Each bounding box $b_i$ contains a set of $M$ tokens transcribed by an OCR engine and coordinates defined as $(x_i^1, x_i^2)$ for the left and right horizontal coordinates and $(y_i^1, y_i^2)$ for the top and bottom vertical coordinates.

## 3 Approach

Our model architecture for visually-asymmetric consistency learning is illustrated in Figure 2. Inspired by mutual learning (Zhang et al., 2018), we start with a standard learning flow and an extra vision-enhanced flow, which are learned simultaneously to transfer knowledge from the vision-enhanced flow to the standard learning flow. It is worth noting that the input images for the vision-enhanced flow are colorful prompt paints, while the input images for the standard flow are original images. Therefore, the information contained in the visual inputs to the vision-enhanced flow and the standard model is asymmetric.

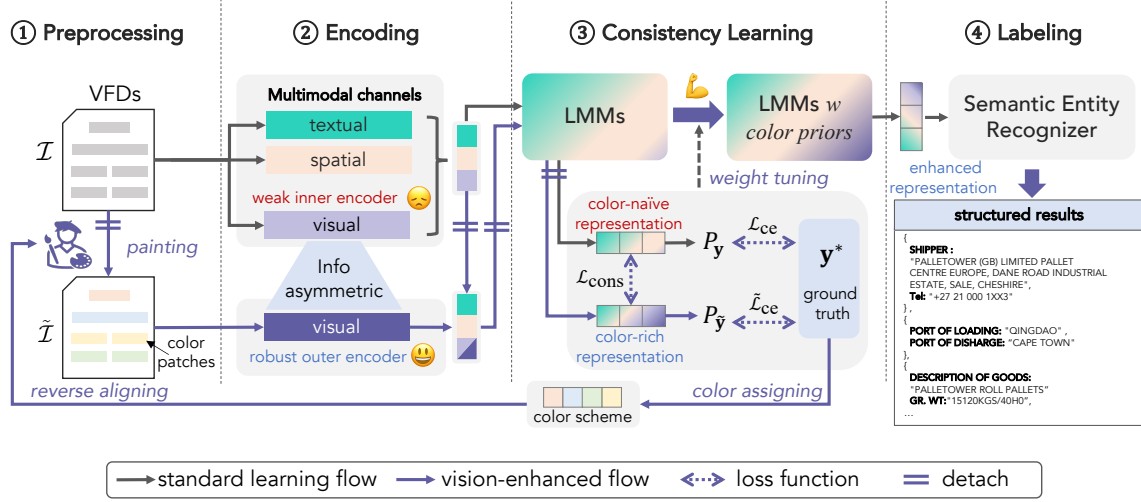

① Preprocessing ② Encoding ③ Consistency Learning ④ Labeling

Figure 2: The overall illustration of the VANCL framework. It encourages knowledge transfer from the extra vision-enhanced flow to the standard flow through consistency losses.

## 3.1 Overall architecture

The proposed approach consists of two entity recognition networks: a Standard Learning (SL) flow to train the backbone LMMs and an extra vision-enhanced (VE) flow to transfer the prior visual knowledge. The two networks have the same structure and share the parameter weights $\Theta$ in the backbone LMMs. Note that the Vision-Enhanced flow has an extra visual encoder, and thus it has additional parameter weights $\Theta_v$ during training. The two flows are logically formulated as follows:

$$P_Y = f_{\text{SL}}(X; \Theta), \quad (1)$$

$$P_{\tilde{Y}} = f_{\text{VE}}(\tilde{X}; \Theta, \Theta_v), \quad (2)$$

where $P_Y$ and $P_{\tilde{Y}}$ are the predicted probability distributions output by the standard learning flow and the extra vision-enhanced flow, which are the latent outputs after softmax normalization (i.e., soft label). Note that the inputs for two networks are different, namely $X$ and $\tilde{X}$, the former is the original document image, and the latter is the synthetic document image (with additional color patches).

The training objective contains two parts: supervision loss $\mathcal{L}_{\text{sup}}$ and consistency loss $\mathcal{L}_{\text{cons}}$. The supervision losses are formulated using the standard cross-entropy loss on the annotated images as follows:

$$\mathcal{L}_{\text{sup}} = \frac{1}{|\mathcal{D}^K|} \sum_{\mathbf{x} \in \mathcal{D}^K} \mathcal{L}_{\text{ce}}\Big( \big( P(\mathbf{y}|\langle \mathcal{B}^k, \mathcal{I}^k \rangle; \Theta), \mathbf{y}^* \big)$$
$$+ \tilde{\mathcal{L}}_{\text{ce}}\big( P(\tilde{\mathbf{y}}|\langle \mathcal{B}^k, \tilde{\mathcal{I}}^k \rangle; \Theta, \Theta_v), \mathbf{y}^* \big) \Big), \quad (3)$$

where $\mathcal{L}_{\text{ce}}$ is the cross-entropy loss function and $\mathbf{y}^*$ is the ground truth. $P(\mathbf{y}|\langle \mathcal{B}^k, \mathcal{I}^k \rangle; \Theta)$ and $P(\tilde{\mathbf{y}}|\langle \mathcal{B}^k, \tilde{\mathcal{I}}^k \rangle; \Theta, \Theta_v)$ refer to the corresponding predicted probability distributions of standard and vision-enhanced models, respectively. $\mathcal{B}^k, \mathcal{I}^k$ denote the bounding box position information and the original image of the $k$-th document. $\tilde{\mathcal{I}}^k$ refers to the synthetic image with color patches.

The consistency loss defines the proximity between two predictions. During training, there exists inconsistency due to asymmetric information in the inputs to the standard learning flow and the vision-enhanced flow. Concretely, it is necessary to penalize the gap between two soft label signals (i.e., the prediction distributions) generated by the standard and vision-enhanced flows. The consistency loss is computed as:

$$\mathcal{L}_{\text{cons}} = \frac{1}{|\mathcal{D}^K|} \sum_{\mathbf{x} \in \mathcal{D}^K} Q\big( P(\mathbf{y}|\langle \mathcal{B}^k, \mathcal{I}^k \rangle; \Theta),$$
$$P(\tilde{\mathbf{y}}|\langle \mathcal{B}^k, \tilde{\mathcal{I}}^k \rangle; \Theta, \Theta_v) \big), \quad (4)$$

where $Q$ is a distance function that measures the divergence between the two distributions.

The final training objective for visually-asymmetric consistency learning is written as:

$$\mathcal{L}_{\text{final}} = \mathcal{L}_{\text{sup}}(\mathbf{y}^*|\Theta, \Theta_v) + \lambda \mathcal{L}_{\text{cons}}(P_{\mathbf{y}}, P_{\tilde{\mathbf{y}}}), \quad (5)$$

where $\lambda$ is a hyperparameter for the trade-off weight. The above loss function takes into account the consistency between hard and soft labels, which also reduces the overconfidence of the model.

## 3.2 Painting with colors

Visual-text alignment is essential for learning multimodal representations, but fine-grained alignment at the bounding box level has not been adequately captured by previous models. Therefore, it is imperative to explore methods for bridging the gap between text segment (or bounding box) detection and entity classification tasks.

One natural solution is to integrate label information using colors, which could effectively enhance visual-text alignment by representing entity-type information with color patches. However, the manual creation of these visual prompts would be extremely time-consuming and laborious. To tackle this problem, we adopt a simple and ingenious process that uses OCR bounding box coordinates to automatically paint the bounding boxes with colors in the original image copies.

Let $\mathcal{D}^k$ denote the $k$-th document, consisting of $\langle \mathcal{B}^k, \mathcal{I}^k \rangle$. First, we make an image copy $\mathcal{I}'^k$ for each training instance of VFD. Then, we paint the bounding boxes in the image copy with the colors responding to entity types according to the coordinates $[(x_i^1, x_i^2), (y_i^1, y_i^2)]$ of bounding box $b_i$. Hence, we obtain a new image $\tilde{\mathcal{I}}^k$ with color patches after painting. This process can be represented as follows:

$$\tilde{\mathcal{I}}^k = \text{Paint}\Big[\text{ROIAlign}\big(\langle \mathcal{B}^k, \mathcal{I}'^k \rangle\big)\Big], \quad (6)$$

where ROIAlign obtains fine-grained image areas corresponding to bounding boxes within the region of interest (here refers to $\mathcal{B}^k$). $\tilde{\mathcal{I}}^k$ is the newly generated synthetic image that preserves the original visual information as well as the bounding box priors with label information. We use these prompt paints to train the extra vision-enhanced flow.

## 3.3 Dual-flow training

Avoiding label leakage becomes a major concern in this task when directly training backbone LMMs with prompt paints using only the standard flow. Fortunately, the dual-flow architecture used in our model not only allows for detaching the vision-enhanced flow and discarding prompt paints when testing but also enables the use of arbitrary backbone LMMs and any available visual encoders in the outer visual channel. This strategy avoids label leakage and enhances the visual feature-capturing ability of the original LMM through a consistency learning-based strategy similar to adversarial learn-

ing. These are exciting and interesting findings in this work.

## 4 Experiments

### 4.1 Datasets

We conduct our experiments on three datasets, FUNSD (Jaume et al., 2019), SROIE (Huang et al., 2019), and a large in-house dataset SEABILL (See Table 1). Details refer to Appendix A.1.

| Dateset | # Train | | | # Test | | |
|---------|---------|------|-------|--------|------|-------|
|         | Doc     | BD   | Token | Doc    | BD   | Token |
| FUNSD   | 149     | 21K  | 33K   | 50     | 8K   | 12K   |
| SROIE   | 626     | 34K  | 155K  | 347    | 19K  | 86K   |
| SEABILL | 3,562   | 250K | 1.5M  | 953    | 74K  | 0.5M  |

Table 1: Statistics of the used datasets, including the numbers of documents (Doc), bounding boxes (BD), and tokens (Token).

### 4.2 Backbone networks

We conduct our experiments using LayoutLM series backbones, a family of transformer-based well-pertained LMMs specifically designed for visually-rich document understanding tasks.

**LAYOUTLM** Vanilla LayoutLM model LAYOUTLM-BASE-UNCASED (Xu et al., 2020) pretrained only using text and layout information.

**LAYOUTLM *w/ img*** Given the vanilla LayoutLM model does not utilize the visual information, we integrate LayoutLM with ResNet-101 (He et al., 2016) to enable the capability of feature extraction from the visual channel[1].

**LAYOUTLMV2/LAYOUTLMV3** We also make comparisons by substituting the backbone with LAYOUTLMV2-BASE-UNCASED (Xu et al., 2021) and LAYOUTLMV3-BASE (Huang et al., 2022).

**VANCL (this work)** We initialize the VANCL from existing pretrained LayoutLM backbones and share the parameter weights of the LayoutLMs in the standard learning and vision-enhanced flows, including the text encoder (e.g., BERT), the inner visual encoder (e.g., ResNet, ResNeXt, and Linear), the position encoders, the Transformer layers, and the classifier. For the outer visual encoder, we use ResNet-101 as the default and also test non-pretrained CNN and ResNet.

---

[1]Note it is different from Xu et al. (2020) in which they use Faster-RCNN (Ren et al., 2015) as the visual encoder.

| Model | #Param | FUNSD | | | SROIE | | | SEABILL | | |
|---|---|---|---|---|---|---|---|---|---|---|
| | | P% ↑ | R% ↑ | F1% ↑ | P% ↑ | R% ↑ | F1% ↑ | P% ↑ | R% ↑ | F1% ↑ |
| BERT[1] | 110M | 54.69 | 61.71 | 60.26 | 90.99 | 90.99 | 90.99 | 66.70 | 68.77 | 67.72 |
| ROBERTA[2] | 125M | 63.49 | 69.75 | 66.48 | 91.07 | 91.07 | 91.07 | 64.35 | 67.76 | 66.01 |
| UNILMv2[3] | 125M | 63.49 | 69.75 | 66.48 | 94.59 | 94.59 | 94.59 | - | - | - |
| BROS[4] | 139M | 80.56 | 81.88 | 81.21 | 94.93 | 96.03 | 95.48 | - | - | - |
| DOCFORMER[5] | 149M | 77.63 | 83.69 | 80.54 | - | - | - | - | - | - |
| LILT[EN-R][6] | - | 87.21 | 89.65 | 88.41 | - | - | - | 84.67 | 86.02 | 85.64 |
| LILT[INFOXLM][6] | - | 84.67 | 87.09 | 85.86 | - | - | - | 85.19 | 87.39 | 86.29 |
| FORMNET[7] | 217M | 85.21 | 84.18 | 84.69 | - | - | - | - | - | - |
| STRUCTEXT[8] | 107M | 85.68 | 80.97 | 83.09 | 95.84 | 98.52 | 96.88 | - | - | - |
| XYLAYOUTLM[9] | - | - | - | 83.35 | - | - | - | 90.75 | 91.59 | 91.17 |
| LAYOUTLM[10] | 113M | 75.97 | 81.55 | 78.66 | 94.38 | 94.38 | 94.38 | 86.93 | 89.16 | 88.03 |
| LAYOUTLM(*w/ img*) | 147M | 79.68 | 80.74 | 80.20 | 95.53 | 96.08 | 95.80 | 87.50 | 89.89 | 88.68 |
| VANCL[LAYOUTLM(*w/ img*)] | +0M | **80.78** | **81.89** | **81.33** | **96.32** | **96.67** | **96.50** | **89.06** | **90.24** | **89.65** |
| LAYOUTLMv2[11] | 200M | 80.29 | **85.19** | 82.76 | 96.25 | 96.25 | 96.25 | 90.90 | 91.73 | 91.31 |
| VANCL[LAYOUTLMv2] | +0M | **82.95** | 83.29 | **83.12** | **97.45** | **97.58** | **97.51** | **91.61** | **92.17** | **91.89** |
| LAYOUTLMv3[12] | 133M | - | - | 90.29 | 96.59 | 96.94 | 96.77 | 89.53 | 91.78 | 90.64 |
| VANCL[LAYOUTLMv3] | +0M | **91.76** | **92.95** | **92.35** | **97.09** | **97.30** | **97.20** | **90.64** | **92.07** | **91.35** |

Table 2: Precision, Recall, and F1 on the FUNSD, SORIE, and SEABILL datasets. [*] denotes the backbone model used in VANCL. **Bold** indicates better scores comparing the model performance between the standard training approach and VANCL. #Param refers to the number of parameters for deployment. [1](Devlin et al., 2019),[2](Liu et al., 2019b),[3](Bao et al., 2020),[4](Hong et al., 2021),[5](Appalaraju et al., 2021),[6](Wang et al., 2022a),[7](Lee et al., 2022),[8](Li et al., 2021b), [9](Gu et al., 2022),[10](Xu et al., 2020),[11](Xu et al., 2021),[12](Huang et al., 2022).

## 4.3 Experimental setups

We train all models based on the default parameters using the Adam optimizer with $\gamma$ of $(0.9, 0.99)$ without warmup. Our models are set to the same learning rate of $5e - 5$. For the FUNSD and SROIE datasets, we set the Dropout to $0.3$ to prevent model overfitting, while we reduce the Dropout to $0.1$ for the SEABILL dataset. We set the training batch size to $8$ and train all models on an NVIDIA RTX3090 GPU. We trained our models for 20 iterations to reach convergence and achieve more stable performance.

## 5 Result Analysis

## 5.1 Overall evaluation

Table 2 shows the scores of precision, recall, and F1 on the form document information extraction task. Our models outperform the baseline models in terms of both LayoutLM (a model that only considers text and spatial features) and LayoutLM *w/ img*/LayoutLMv2/LayoutLMv3 (models that also consider visual features). It should be noted that LayoutLM *w/ img* incorporates visual features after the spatial-aware multimodal transformer layers, while LayoutLMv2 and LayoutLMv3 incorporate visual features before these layers. It suggests that VANCL could be easily applied to most of the ex-

isting LMMs for visually-rich document analysis and information extraction tasks with little or no significant modifications to the network architecture. Please refer to Appendix A.3 for a detailed case study.

## 5.2 Impact of consistency loss

As for a comprehensive evaluation of each module, we conduct the additional ablation experiment by examining the impact of using consistency loss. The experimental results in Table 3 reveal that removing the consistency loss leads to a decline in the model's performance to varying extents. This find-

| Model | FUNSD | SROIE | SEABILL |
|---|---|---|---|
| LAYOUTLM(*w/ img*) | | | |
| + VANCL | **81.33** | **96.50** | **89.65** |
| + VANCL-CL | 79.36(↓ 1.97) | 95.68(↓ 0.82) | 88.19(↓ 1.46) |
| LAYOUTLMv2 | | | |
| + VANCL | **83.12** | **97.51** | **91.89** |
| + VANCL-CL | 81.52(↓ 1.60) | 97.08(↓ 0.43) | 89.28(↓ 2.61) |
| LAYOUTLMv3 | | | |
| + VANCL | **92.35** | **97.20** | **91.35** |
| + VANCL-CL | 90.93(↓ 1.42) | 96.98(↓ 0.22) | 90.87(↓ 0.48) |

Table 3: Ablation experiment by examining the impact of using consistency loss on different backbones. - CL means no consistency loss is used.

ing demonstrates the significance of consistency loss in the model. Simultaneously, it indicates that

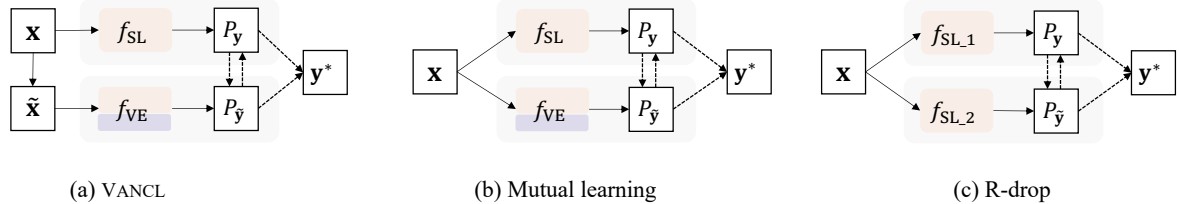

| (a) VANCL | (b) Mutual learning | (c) R-drop |

Figure 3: Illustrating the architectures for (a) our approach VANCL, (b) mutual learning, (c) R-drop. '→' means forward operation and '- ->' means loss of supervision.

without the driving force of consistency, the addition of color information may actually increase the learning noise.

## 5.3 Different divergence metrics

There are many ways to measure the gap between two distributions, and different measures can lead to different consistency loss types, which can also affect the final performance of the model on form document extraction tasks. In this section, we verify two types of consistency loss, Jensen-Shannon (JS) divergence (Lin, 1991) and Kullback-Leibler (KL) divergence (Kullback and Leibler, 1951), on the effect of form document extraction. While the

| Model | FUNSD | SROIE | SEABILL |
|---|---|---|---|
| LAYOUTLM(*w/ img*) | 80.20 | 95.80 | 88.68 |
| +VANCL-JS | 80.75 | **96.50** | 89.36 |
| +VANCL-KL | **81.33** | 96.32 | **89.65** |
| LAYOUTLMV2 | 82.76 | 96.25 | 91.31 |
| +VANCL-JS | 82.21 | 97.33 | **91.89** |
| +VANCL-KL | **83.12** | **97.51** | 91.58 |
| LAYOUTLMV3 | 90.29 | 96.77 | 90.64 |
| +VANCL-JS | **92.35** | 97.07 | 91.16 |
| +VANCL-KL | 91.72 | **97.20** | **91.35** |

Table 4: Effect of different consistency losses using JS and KL divergences on F1 scores. **Bold** indicates the best and underline indicates the second best.

VANCL model outperforms the baseline in most cases, regardless of the different backbone networks and datasets, it is still worth noting that the choice of consistency loss varies depending on the characteristics of the dataset, such as key type and layout style, and whether overfitting occurs due to the complexity of the model and the data size. As shown in Table 4, different consistency losses are used for different datasets and backbone networks to achieve optimal results. For example, when using Jensen-Shannon divergence for LayoutLMv2, VANCL achieved optimal results on the SEABILL dataset, but second best on the FUNSD and SROIE

datasets.

## 5.4 Comparison with inner/outer consistency

To demonstrate the effectiveness of our proposed method, we compare our approach versus R-Drop (Liang et al., 2021) and mutual learning (Zhang et al., 2018). R-Drop is a powerful and widely used regularized dropout approach that considers multiple model inner consistency signals. Figure 3 illustrates the different structures of the three methods. Table 5 gives the results of incorporating LayoutLMs with R-Drop and mutual learning. Compared to R-Drop and mutual learning, we observe a positive correlation between using VANCL and the improvements on the FUNSD, SROIE, and SEABILL datasets. The results indicate that VANCL benefits from the vision priors.

| Model | FUNSD | SROIE | SEABILL |
|---|---|---|---|
| LAYOUTLM (*w/ img*) | 80.20 | 95.80 | 88.68 |
| +MUTUAL-LEARN | 80.44 | 95.46 | 88.99 |
| +R-DROP | 80.38 | 96.02 | 89.13 |
| +VANCL | **81.33** | **96.50** | **89.65** |

Table 5: Comparison of F1 scores with inner and outer consistencies using R-Drop (Liang et al., 2021) and mutual learning (Zhang et al., 2018).

## 5.5 Visualization analysis

To more intuitively demonstrate the effect of VANCL on the obtained visual encodings in the visual channel, we visualize the encoding distributions before and after VANCL training via t-SNE. Figure 4 shows the integration of visual information in both flows. This demonstrates that VANCL indirectly transfers the label prior distribution to the standard flow by aligning the information in the standard and vision-enhanced flows, which improves the subsequent inference process. More detailed results of the overall distribution visualization can be found in Appendix A.4.

| | QUESTION | | ANSWER | | HEADER | | OTHER | | MICRO-AVG |
|---|---|---|---|---|---|---|---|---|---|
| | color | F1 | color | F1 | color | F1 | color | F1 | |
| 1 | #FF0000 | 83.88 | #0000FF | **85.51** | #00FF00 | 56.31 | #FFA500 | 77.36 | **81.33** |
| 2 | #0000FF | **84.42** | #FF0000 | 85.49 | #00FF00 | 57.92 | #FFA500 | 75.83 | 81.28 |
| 3 | #FFA500 | 84.01 | #00FF00 | 85.22 | #FF0000 | 54.08 | #0000FF | 77.38 | 81.27 |
| 4 | #CCCCCC | 83.67 | #999999 | 84.78 | #333333 | 57.18 | #000000 | **77.47** | 81.16 |
| 5 | #FF0000 | 83.52 | #FF6699 | 84.88 | #FF3366 | 56.02 | #FF0099 | 76.23 | 80.75 |
| 6 | #0000FF | 83.98 | #FF0000 | 84.59 | #0099FF | 56.40 | #0066CC | 76.42 | 80.76 |
| 7 | #FF0000 | 83.43 | #0000FF | 83.62 | #FFA500 | 55.73 | #00FF00 | 76.30 | 80.23 |
| 8 | #FFFFFF | 83.45 | #FFFFFF | 84.21 | #FFFFFF | **61.40** | #FFFFFF | 75.58 | 80.38 |
| SUPP. | 2,542 | | 3,294 | | 374 | | 2,356 | | 8,566 |

Table 6: Results of using different color schemes on the FUNSD dataset. SUPP. denotes the number of supports for each entity type in the test set.

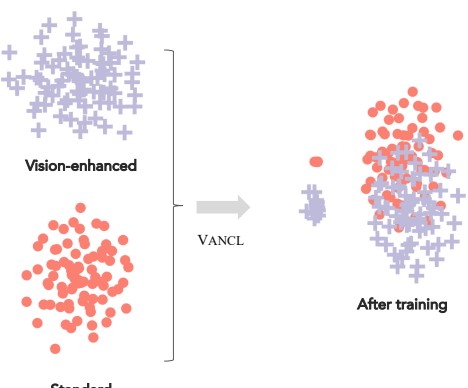

Figure 4: The t-SNE visualization of token-level visual encodings of a particular type output by the standard (red) and vision-enhanced (grayish purple) flows.

## 5.6 Changing the color scheme

To verify the effect of different background colors on the model, we conduct experiments on the FUNSD dataset. In addition to the standard color scheme (red, blue, yellow, and green), we conduct three additional experiments. First, we randomly swap the colors used in the standard color scheme. Second, we choose different shades or intensities of the same color system for each label class. Third, we only draw the bounding box lines with colors. The results in Table 6 reveal three rules followed by VANCL, which are consistent with the existing findings in cognitive science (Gegenfurtner, 2003; Elliot and Maier, 2014): 1) different shades or intensities of the same color system do not significantly affect the results; 2) swapping different colors in the current color scheme does not significantly affect the results; and 3) VANCL is effective when using colors with strong contrasts while being sensitive to grayscale.

| Model | Ratio (F1) | | | | |
|---|---|---|---|---|---|
| | 5% | 12.5% | 25% | 50% | 100% |
| LAYOUTLM(*w/ img*) | 71.74 | 80.67 | 83.72 | 85.97 | 88.68 |
| +VANCL | **76.33** | **82.09** | **84.83** | **87.14** | **89.65** |
| LAYOUTLMv2 | 80.63 | 84.48 | 88.12 | 89.47 | 91.31 |
| +VANCL | **84.75** | **87.42** | **89.34** | **91.18** | **91.89** |

Table 7: The impact of reducing the data size of the training subset on the SEABILL dataset.

## 5.7 Low-resource scenario

To verify the effectiveness of VANCL in low-resource scenarios, we investigate whether VANCL improves the SER task when training the model with different sizes of data. We choose LayoutLM and LayoutLMv2 as the test backbone and compare the results of VANCL with the corresponding baseline by varying the training data sizes. Specifically, we randomly draw a percentage $p$ of the training data, where $p$ is chosen from the set $\{5\%, 12.5\%, 25\%, 50\%\}$, from the SEABILL dataset. Results in Table 7 reveal two key observations. First, VANCL consistently outperforms the LayoutLM baselines for different sizes of training data. Second, the performance gap between VANCL and the baseline is much larger in a low-resource scenario, indicating that VANCL can boost the model training in such scenarios.

## 5.8 Non-pretrained testing

To verify whether VANCL's improvement is simply due to pretrained visual encoders containing real-world color priors, we deliberately tested visual encoders that were not pretrained and initialized their parameters when introduced into the model. In this way, the visual encoder would need to learn to extract visual features from scratch. Considering

| Model | Enc. | #L | Pre. | FUNSD | SEABILL |
|-------|------|----|----|-------|---------|
| STAND. | - | - | - | 78.66 | 88.03 |
| STAND. | RESNET | 101 | ✔ | 80.20 | 88.68 |
| STAND. | RESNET | 18 | ✘ | 79.51 | 88.50 |
| STAND. | CNN | 2 | ✘ | 80.03 | 88.42 |
| VANCL | RESNET | 101 | ✔ | **81.33** | **89.65** |
| VANCL | RESNET | 18 | ✘ | 77.62 | 88.64 |
| VANCL | CNN | 2 | ✘ | 80.60 | 88.97 |

Table 8: Effect of using a non-pretrained outer visual encoder on LAYOUTLM (*w/img*). Pre. means pretrained. Enc. and #L denote the outer visual encoder and the number of inside layers.

that deeper networks are usually more difficult to train in a short time, we choose a smaller ResNet-18 and an extremely simple two-layer CNN (Lecun et al., 1998) as the outer visual encoders for our experiments. The results in Table 8 show that even simple visual encoder networks, such as CNN-2 and ResNet-18, can still surpass the standard baseline models, indicating that the original backbone models learn better visual features through visually-asymmetric consistency learning (VANCL).

# 6 Related Work

## 6.1 Visually-rich document understanding

To achieve the goal of understanding visually-rich documents, a natural attempt is to enrich state-of-the-art pretrained language representations by making full use of the information from multiple modalities including text, layout and visual features. Researchers have proposed various paradigms, such as text-based, grid-based, graph-based, and transformer-based methods. **Text-based methods**, e.g., XLM-RoBERT (Conneau et al., 2020), InfoXLM (Chi et al., 2021), only consider the textual modality, rely on the representation ability of large-scaled pretrained language models. **Grid-based methods**, e.g., Chargrid (Katti et al., 2018), BertGrid (Denk and Reisswig, 2019), and ViBERTgrid (Lin et al., 2021), represent documents using a 2D feature map, allowing for the use of image segmentation and/or object detection models in computer vision (CV). **GNN-based methods** (Liu et al., 2019a; Tang et al., 2021) treat text segments in the document as nodes and model the associations among text segments using graph neural networks. **Transformer-based methods** leverage the latest multimodal pretrained models to learn better semantic representations for VFDs by capturing in-

formation from multiple modalities (Powalski et al., 2021; Wang et al., 2022b). LayoutLM (Xu et al., 2020) introduces 2D relative position information based on BERT (Devlin et al., 2019), which enables the model to perceive the 2D positions of text segments throughout the document. LayoutLMv2 (Xu et al., 2021), StructText (Li et al., 2021b), Struct-Textv2 (Yu et al., 2023) and LayoutLMv3 (Huang et al., 2022) further integrate visual channel input into a unified multimodal transformer framework to fusing textual, visual, and spatial features. Thus, these methods learn more robust multimodal representations and have made promising progress in visually rich document understanding tasks. Recently, Wang et al. (2022a) recently propose a bi-directional attention complementation mechanism (BiACM) to enable cross-modality interaction independent of the inner language model. Lee et al. (2022) exploit serializing orders of the text segment output by OCR engines.

## 6.2 Consistency learning

In recent years, there have been significant advancements in consistency learning, which aims to reduce variances across different model predictions. The core idea of consistency learning is to construct cross-view constraints that enhance consistency across different model predictions. Previous research on consistency learning has focused on adversarial or semi-supervised learning, which provides supervised signals that help models learn from unlabeled data (Miyato et al., 2019; Xie et al., 2020; Lowell et al., 2021a). There is also interest in incorporating consistency mechanisms into supervised learning (Chen et al., 2021a; Lowell et al., 2021b). Batra and Parikh (2017) propose cooperative learning, allowing multiple networks to learn the same semantic concept in different environments using different data sources, which is resistant to semantic drift. Zhang et al. (2018) propose a deep mutual learning strategy inspired by cooperative learning, which enables multiple networks to learn from each other.

Previous works enable cross-modal learning either using the classical knowledge distillation methods (Hinton et al., 2015) or modal-deficient generative adversarial networks (Ren and Xiong, 2021). Hinton et al. (2015)'s method starts with a large, powerful pretrained teacher network and performs one-way knowledge transfer to a simple student, while Ren and Xiong (2021)'s method attempts to

encourage the partial-multimodal flow as strong as the full-multimodal flow. In contrast, we explore the potential of cross-modal learning and information transfer, and enhance the model's ability to information extraction by using visual clues. This allows the model to learn a priori knowledge of entity types, which is acquired before encountering the specific task or data. In terms of model structure, our model architecture is more similar to mutual learning (Zhang et al., 2018) or cooperative learning (Batra and Parikh, 2017).

## 7    Conclusion

We present VANCL, a novel consistency learning approach to enhance layout-aware pretrained multimodal models for visually-rich form-like document information extraction. Experimental results show that VANCL successfully learns prior knowledge during the training phase, surpassing existing state-of-the-art models on benchmark datasets of varying sizes and a large-scale real-world form document dataset. VANCL exhibits superior performance against strong baselines with no increase in model size. We also provide recommendations for selecting color schemes to optimize performance.

## Limitations

The limitations of our work in this paper are the following points:

1. For the sake of simplicity, we only experiment with the LayoutLM-BASE, LayoutLMv2-BASE, and LayoutLMv3-BASE backbones, we have not tested other backbones, e.g., StrucText and StrucTextv2 .

2. We have not studied the effectiveness of asymmetric consistency learning in the training stage.

## Acknowledgements

This work was supported by National Natural Science Foundation of China (Young Program: 62306173, General Program: 62176153), JSPS KAKENHI Program (JP23H03454), Shanghai Sailing Program (21YF1413900), Shanghai Pujiang Program (21PJ1406800), Shanghai Municipal Science and Technology Major Project (2021SHZDZX0102), the Alibaba-AIR Program (22088682), and the Tencent AI Lab Fund (RBFR2023012).

## Ethical Statements

The potential ethical implications of this work are discussed here. (1) **Data**: FUNSD and SROIE datasets are publicly available, accessible, and well-developed benchmarks for document understanding. (2) **Privacy-related**: The SEABILL dataset is collected from a real-world business scenario. While the data provider has deleted sensitive information, such as personally identifiable information, the SEABILL dataset still contains some location names and company names. Given the above reason, we could not release this dataset. (3) **Reproducibility**: Our model is built on open-source models and codes, and we will release the code and model checkpoints later to reproduce the reported results. (4) **Ethical issues**: The proposed approach itself will not produce information that harms anyone. Therefore, we do not anticipate any ethical issues.

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

# A  Appendix

## A.1  Dataset Details

**FUNSD** This dataset contains 199 well-annotated scanned forms. Each semantic entity consists of a unique identifier id, a label ({*Question, Answer, Header, or Other*}), a bounding box, a list of links to other entities, and a list of words.

**SROIE** The dataset contains 626 receipts for training and 347 receipts for testing. Each receipt consists of its scanned image and OCR transcription, organized as a list of text segments with bounding box position information pairs. Each receipt is labeled with four types of entities, namely {*Company, Date, Address, Total*}.

**SEABILL** The dataset is a complex collection of documents derived from maritime business scenarios and consists of 3,562 training documents and 953 test documents. The data consists of PDF images and rule-based transcriptions of the documents with three labels {*Question, Answer, Other*}.

## A.2  Effect of sharing weights

In this section, we design experiments to investigate the effect of sharing weight. As shown in Table 9, we conduct experiments on three datasets by controlling the variables of the inputs and model weight sharing.

| Visual | Prompt | FUNSD | SROIE | SEABILL |
|--------|:------:|-------|-------|---------|
| ¬ *shared* | ✘ | 77.47 | 95.13 | 88.63 |
|  | ✔ | 80.03 | 95.84 | 89.13 |
| *shared* | ✘ | 80.38 | 96.02 | 89.13 |
|  | ✔ | **81.33** | **96.50** | **89.65** |

Table 9: Effect of sharing weights of the network parameters between the inner and outer visual encoders in LAYOUTLM (*w/ img*) on model F1 score. Sharing the weights of the model brings significantly better results than not sharing the weights.

It can be seen that sharing the parameters of the visual encoder is better than not sharing the weights in most cases (see Table 9), and the way of sharing parameters also greatly reduces the overall number of parameters of the model (see Table 2). In addition, we also compare the results whether using the colorful prompt paints under the settings of weight sharing or not. Even under the condition that model weights are not shared, we observe that colored visual clues still significantly improve the performance of the model.

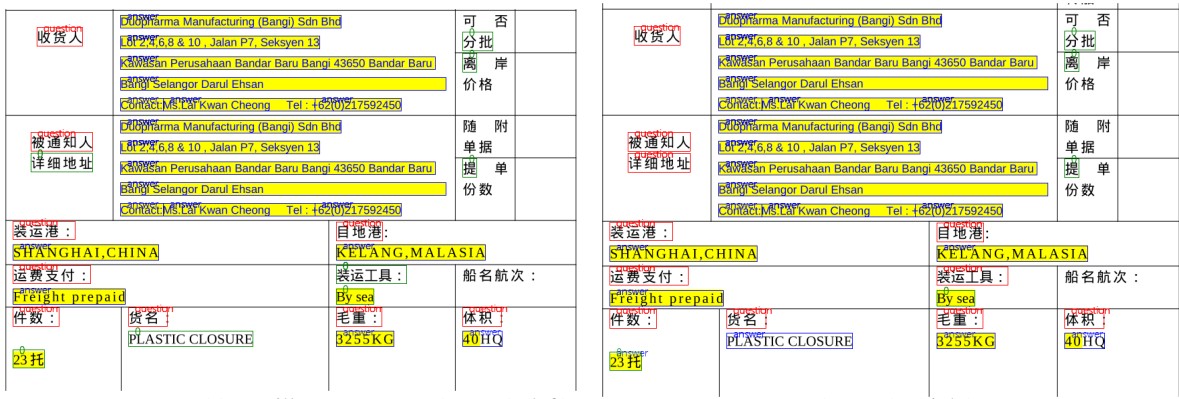

(a) Vanilla LAYOUTLM(*w/ img*) (left) vs. VANCL[LAYOUTLM(*w/ img*)] (right)

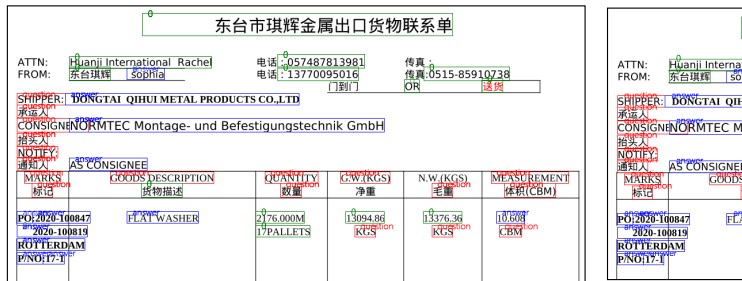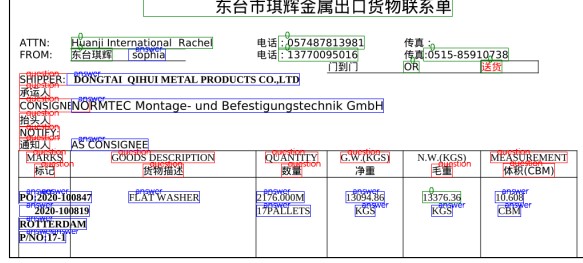

(b) Vanilla LAYOUTLMV3 (left) vs. VANCL[LAYOUTLMV3] (right)

Figure 5: Case study of the predicted results on the SEABILL dataset using the standard fine-tuning pipeline LAYOUTLM(*w/ img*) (a) and LAYOUTLMV3 against the VANCL pipeline.

## A.3 Case study

Figure 5(a) and Figure 5(b) visualize the corresponding prediction results using the vanilla LAYOUTLM(*w/ img*) and LAYOUTLMV3 models against VANCL[LAYOUTLM(*w/ img*)] and VANCL[LAYOUTLMV3] models on the SEABILL dataset, respectively. In Figure 5(a), LAYOUTLM(*w/ img*) predicts wrong label "*Other*' for "PLASTIC CLOSURE", "装运工具 (Loading way)", "详细地址 (Detailed address)", "23托 (23 GP)", while VANCL[LAYOUTLM(*w/ img*)] can make correct predictions that match the human-annotated ground truths. There are ambiguous cases that VANCL[LAYOUTLM(*w/ img*)] also gives wrong predictions. For example, both LAYOUTLM(*w/ img*) and VANCL[LAYOUTLM(*w/ img*)] predict "By sea (23 GP)" as "*Other*", but the true label is "*Answer*".

Figure 5(b) shows that LAYOUTLMV3 gives more accurate predictions than LAYOUTLM(*w/ img*). However, LAYOUTLMV3 also predicts the entities with the wrong label "*Other*" for "货物描述 (Goods description)", "2176.000M", "17PAL-LETS", "13094.86" though these entities are annotated as "*Question*" or "*Answer*" in the training data. For "KGS", "CBM", LAYOUTLMV3

also gives a wrong label "*Question*", whereas VANCL[LAYOUTLMV3] gives the right label.

## A.4 Visualization

The full version of the t-SNE visualization is shown in Figure 6. The upper left and upper right graphs show a comparison of the visual encodings of the two flows before and after VANCL training, respectively. After training, the results show that each classification cluster is more separated and the corresponding information of the two flows is more aligned. The lower left and lower right graphs show a comparison of the multimodal output of the two flows before and after VANCL training, respectively. In summary, VANCL effectively transfers label information to the standard flow through the vision-enhanced flow.

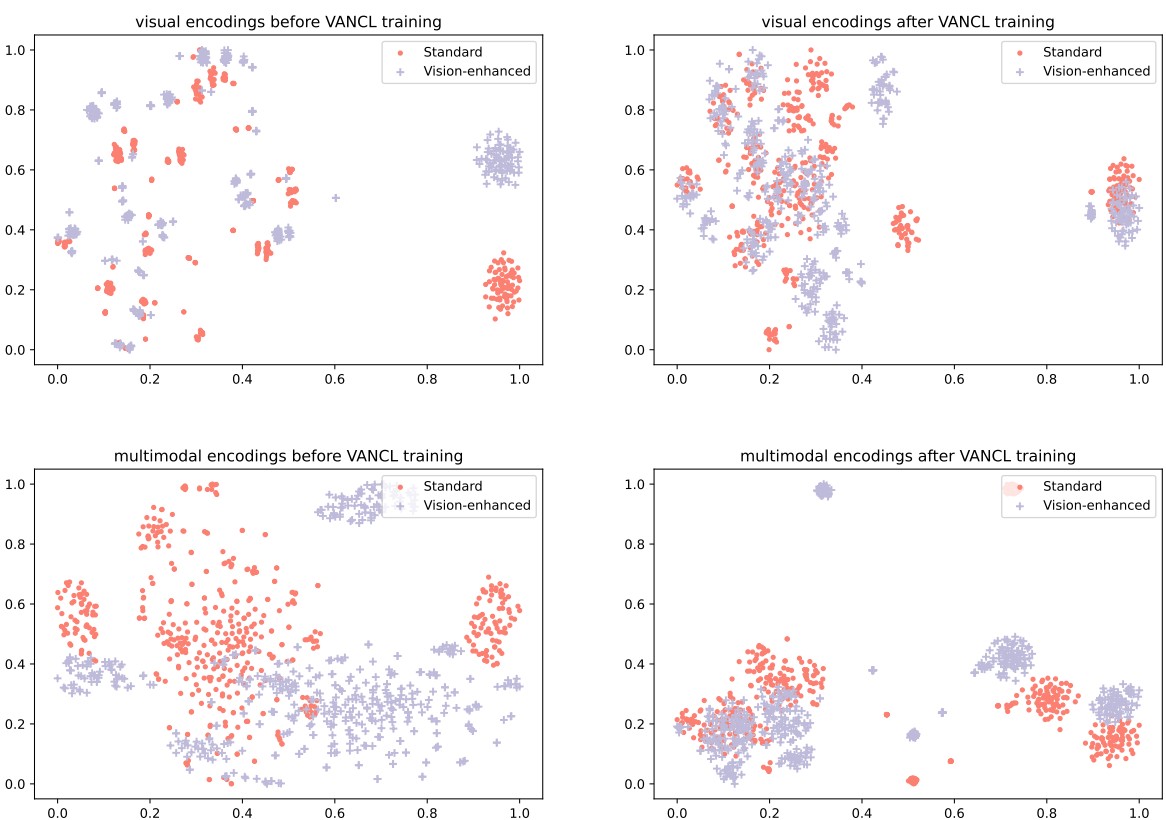

Figure 6: The full t-SNE visualization result of hidden states from the outputs of the standard and vision-enhanced flows. The uppers are visual encodings before and after VANCL training. The lowers are multimodal fused encodings before and after VANCL training. The red bounding boxes indicate the clusters shown in Section 5.5.