# OpenReview forum: "Vision-Enhanced Semantic Entity Recognition in Document Images via Visually-Asymmetric Consistency Learning"
_EMNLP/2023/Conference — EMNLP 2023 Main_

### Official Review · Reviewer_XaQk · 2023-08-03

**Soundness:** 4

**Excitement:**

4: Strong: This paper deepens the understanding of some phenomenon or lowers the barriers to an existing research direction.

**Paper Topic And Main Contributions:**

This paper presents a method for semantic entity recognition, in which some entities in a document (e.g., headers, questions, and answers) are identified. The main idea of the paper is to facilitate training a model for this task with additional visual input that encodes the semantic entity types using colors. The paper also proposes a training scheme with this additional visual input.

**Questions For The Authors:**

(A) I would like to have a discussion on the effectiveness of consistency loss.
(B) It would be nice to have some discussions on the (not necessarily statistic) significance of the performance gain.

**Reasons To Accept:**

The paper is well-written and the method is well-motivated. Although the performance gain by the method looks marginal, I think the idea is interesting. Also, the paper provides a good set of comparisons and ablation studies (though some aspects are missing).

**Reasons To Reject:**

As far as I understood, the ablation study over the consistency loss (whether it’s used or not) is not provided. This point is the main drawback of the paper.

The performance gain is marginal. Together with the above point, this makes the evaluation of the method difficult.

**Reproducibility:**

4: Could mostly reproduce the results, but there may be some variation because of sample variance or minor variations in their interpretation of the protocol or method.

**Reviewer Confidence:**

3: Pretty sure, but there's a chance I missed something. Although I have a good feel for this area in general, I did not carefully check the paper's details, e.g., the math, experimental design, or novelty.

---

> ### Author Rebuttal · Authors · 2023-08-28
>
> Thank you very much for your suggestion for an ablation experiment!
>
> ***Response-Q1: Ablation study over the consistency loss***
>
> As the reviewers' suggested, we have now included an ablation study to investigate the impact of utilizing consistency loss. We believe now it is sufficient to support the initial motivation.
>
> The experimental results are presented below:
>
>
> | Model                           | FUNSD（F1-score）        | SROIE （F1-score）       | SEABILL  （F1-score）    |
> | ------------------------------- | ------------ | ------------ | ------------ |
> | LayoutLM                        | 80.20        | 95.80        | 88.68        |
> | VANCL [LayoutLM]                | 81.33        | 96.50        | 89.65        |
> | VANCL [LayoutLM] -consis_loss   | **79.36(-1.97)** | **95.68(-0.82)** | **88.19(-1.46)** |
> | -                               | -            | -            | -            |
> | LayoutLMv2                      | 82.76        | 96.25        | 91.31        |
> | VANCL [LayoutLMv2]              | 83.12        | 97.50        | 91.89        |
> | VANCL [LayoutLMv2] -consis_loss | **81.52(-1.60)** | **97.08(-0.42)** | **89.28(-2.61)** |
> | -                               | -                | -                | -                |
> | LayoutLMv3                      | 90.29            | 96.77            | 90.64             |
> | VANCL [LayoutLMv3]              | 92.35            | 97.20            | 91.35            |
> | VANCL [LayoutLMv3] -consis_loss | **90.93(-1.42)** | **96.98(-0.22)** | **90.87(-0.48)** |
>
> The performance of the model has experienced varying degrees of decline after removing the consistency loss. On one hand, this highlights the significance of consistency loss. On the other hand, it demonstrates that without the consistency loss, the additional color information actually introduces learning noise. This implies that merely incorporating colors is not sufficient; the process of consistency learning is essential. This finding further supports the conclusion drawn in Section 5.5 from another perspective: the color itself is not our primary concern.
>
> ***Response-Q2: Some discussions on the significance of the performance gain***
>
> While we do not explicitly present statistical significance, we give case study and visualization to better support the performance gain, which offers a reliable indication of the validity of our findings.
>
> Case Study: Appendix A.3 provides several examples for comparing results, and we can offer additional ones if necessary.
>
> Visualization: Refer to Figure 4 and Appendix A.4. Through consistency learning, the visual representation of the standard flow becomes more similar to that of the vision-enhanced flow, ultimately making the multi-modal representation more effective. Since we know that the vision-enhanced flow representation is robust, we can also consider the standard flow representation to be enhanced when it aligns with the vision-enhanced representation.

---

### Official Review · Reviewer_XvBS · 2023-08-04

**Typos Grammar Style And Presentation Improvements:** The authors could change the color of…
**Soundness:** 2

**Excitement:**

3: Ambivalent: It has merits (e.g., it reports state-of-the-art results, the idea is nice), but there are key weaknesses (e.g., it describes incremental work), and it can significantly benefit from another round of revision. However, I won't object to accepting it if my co-reviewers champion it.

**Paper Topic And Main Contributions:**

This paper proposes VANCL to improve visual encoders of LMM. Specifically, it utilizes asymmetric consistency learning to enhance the visual representations in the visual encoder. With no extra cost during inference time, VANCL could achieve better results then without the visual enhanced flow.

**Questions For The Authors:**

Does all LMMs have  weaker representational capabilities in visual encoders?
I cannot understand Figure 4. Can you help explain why after training, the representation is vision enhanced?
Can the authors give some explanations on of JS and KL in consistency loss?

**Reasons To Accept:**

1. The results are impressive without extra cost in inference. The paper potentially provides a general way to improve complex document understanding.
2. The findings are consistent with psychology theory, and the color schemes are useful to future research in complex document understanding.
3. Overall, the presentation is clear and easy to understand.

**Reasons To Reject:**

1. The authors point out that visual encoders in previous LMMs have weaker representational capabilities in L72-74. It's not convincing as there is no evidence to show this.
2. The results cannot achieve state-of-the-art results with LayoutLM. I would suggest authors producing more results with LILT and FORMNET backbones.

**Reproducibility:**

3: Could reproduce the results with some difficulty. The settings of parameters are underspecified or subjectively determined; the training/evaluation data are not widely available.

**Reviewer Confidence:**

3: Pretty sure, but there's a chance I missed something. Although I have a good feel for this area in general, I did not carefully check the paper's details, e.g., the math, experimental design, or novelty.

---

> ### Author Rebuttal · Authors · 2023-08-28
>
> Thank you for your valuable feedback!
>
> ***Response-Q1: "Previous LMMs have weaker representational capabilities ... no evidence to show this.***
>
> Sorry for the confusion.
> We want to clarify that the intended meaning of this statement is that the visual encoders in earlier LMMs are somewhat less effective than the most recent OCR engines. This is due to the absence of task-specific visual features in those encoders, such as text segment detection and bounding box regression.
>
> In reality, we do not explicitly state that the visual encoders of previous LMMs are weak; rather, we emphasize that their visual representations in VFD-specific domains are biased. General visual representation alone is insufficient for VFD information extraction. In this work, we enforce the model to extract task-specific visual features.
>
> ***Response-Q2: SOTA and other backbones***
>
> In fact, we believe that we have achieved state-of-the-art (SOTA) results. The fact has been confirmed by all the previous round of ACL reviewers. However, we recently found FormNetV2 (ACL2023), a concurrent work, but comparing with our method, it only achieves better scores on the SRIOE dataset.
>
> To our knowledge, FormNetV1 does not effectively utilize visual features, while LILT primarily addresses cross-language issues. They are not suitable to be a backbone model. However, it might be worthwhile to explore extensions to FormNetV2 in the future – a promising suggestion.
>
> We would like to caution reviewers to concentrate more on the problem our paper addresses rather than solely focusing on the resulting SOTA.
>
> ***Response-Q3:why after training, the representation is vision enhanced?***
> As previously elucidated, the visual representation within the specific domain of VFD is relatively weak, and the general visual representation does not sufficiently fulfill the requirements for information extraction in the context of VFD.
>
> In reference to Figure 4 and Appendix A.4, the intention is not to directly augment the visual representation. Rather, through the implementation of consistency learning, the standard visual representation and the vision-enhanced flow representation become more congruent, ultimately rendering the multimodal representation more efficacious. Given that the visual representation of the vision-enhanced flow is robust, it can be inferred that if the standard flow representation converges towards this vision-enhanced representation, the standard flow representation can also be considered enhanced.
>
> ***Response-Q4:KL and JS?***
> See line 193-194 and Equation.4, KL divergence is used to measure the degree of difference between two probability distributions, the larger the KL divergence, the greater the difference between the two; when the KL divergence is smaller, it means that the difference between the two is smaller. The difference between JS divergence and KL divergence is that KL divergence is asymmetric, while JS divergence is symmetric.
>
> ***Response-Q5: Presentation Improvement: "... change the color of Figure2. ...hard to read."***
>
> Thank you very much for your feedback.
> We will change the color. If there are any areas that are difficult to read or require further clarification, please do not hesitate to point them out, and we will try to improve the presentation.

---

### Official Review · Reviewer_uW1H · 2023-08-09

**Soundness:** 3

**Excitement:**

4: Strong: This paper deepens the understanding of some phenomenon or lowers the barriers to an existing research direction.

**Paper Topic And Main Contributions:**

The paper proposes a model for enhancing visual encoding, which demonstrates remarkable performance in table recognition.

**Reasons To Accept:**

1. The motivation behind this article is highly reasonable, as colors evidently have a positive impact on human recognition of table content in reality.
2. The model design is sound.
3. We observe a significant increase in performance.

**Reasons To Reject:**

1. The authors should provide more analysis on why the method proposed in this article can more effectively utilize color information.
2. The ablation experiments are insufficient; it is strongly recommended that the authors conduct a comprehensive evaluation of each module.

**Reproducibility:**

4: Could mostly reproduce the results, but there may be some variation because of sample variance or minor variations in their interpretation of the protocol or method.

**Reviewer Confidence:**

3: Pretty sure, but there's a chance I missed something. Although I have a good feel for this area in general, I did not carefully check the paper's details, e.g., the math, experimental design, or novelty.

---

> ### Author Rebuttal · Authors · 2023-08-28
>
> Thanks for your valuable comments!
>
> ***Response-Q1:"Why the proposed method utilize color information more effectively ."***
>
> Colors are primarily incorporated into the fine-tuning process as a prior knowledge of entity-level labels, as described in Section 3.2. In fact, color information simply serves as a mapping of the label for the vision channel.
>
> In summary, the primary objective of this paper is to enhance the multimodal output of the standard flow model by reducing the divegence between them. We utilize color information as a means to improve the model by striving to achieve consistency between the output of the standard model and the output of the visual enhancement through consistency learning.
>
>
>
>
> ***Response-Q2: Ablation Experiments***
>
> Our model primarily contains on two modules: color painting and consistency learning. The ablation experiment for color painting involves ablating the color, which includes scenarios such as no color, different colors, and varying shades of colors. Please refer to Section 5.5 for more details on this ablation experiment.
>
> As for a comprehensive evaluation of each module, we have conducted the additional ablation experiment by examining the impact of using consistency loss. The experimental results reveal that removing the consistency loss leads to a decline in the model's performance to varying extents. This finding demonstrates the significance of consistency loss in the model. Simultaneously, it indicates that without the driving force of consistency, the addition of color information may actually increase the learning noise.
>
>
> | Model                           | FUNSD（F1-score）             | SROIE （F1-score）      | SEABILL （F1-score）    |
> | ------------------------------- | ---------------- | ---------------- | ---------------- |
> | LayoutLM                        | 80.20            | 95.80            | 88.68            |
> | VANCL [LayoutLM]                | 81.33            | 96.50            | 89.65            |
> | VANCL [LayoutLM] -consis_loss   | **79.36(-1.97)** | **95.68(-0.82)** | **88.19(-1.46)** |
> | -                               | -                | -                | -                |
> | LayoutLMv2                      | 82.76            | 96.25            | 91.31            |
> | VANCL [LayoutLMv2]              | 83.12            | 97.50            | 91.89            |
> | VANCL [LayoutLMv2] -consis_loss | **81.52(-1.60)** | **97.08(-0.42)** | **89.28(-2.61)** |
> | -                               | -                | -                | -                |
> | LayoutLMv3                      | 90.29            | 96.77            | 90.64             |
> | VANCL [LayoutLMv3]              | 92.35            | 97.20            | 91.35            |
> | VANCL [LayoutLMv3] -consis_loss | **90.93(-1.42)** | **96.98(-0.22)** | **90.87(-0.48)** |
>
>
> Furthermore, we kindly suggest reviewing Figure 4 and Appendix A.4 for a more comprehensive understanding of our research. These sections primarily demonstrate that, by employing consistency learning, the multimodal output of the standard flow tends to lean towards the vision-enhanced flow. This, in turn, facilitates the learning of the mapping relationship between the entity and its corresponding ground truth label.

---

### Official Review · Reviewer_SYbT · 2023-08-12

**Soundness:** 5

**Excitement:**

4: Strong: This paper deepens the understanding of some phenomenon or lowers the barriers to an existing research direction.

**Paper Topic And Main Contributions:**

The paper tackles the problem of multimodal information extraction, specifically entity recognition from visually rich form-like documents.

The authors argue that existing multimodal extraction systems like LayoutLM (referred to as large scale pretrained multimodal models, LMMs) are limited in leveraging fine grained visual features, and propose a novel objective called visually asymmetric consistency learning (VANCL) that can be used with any multimodal encoder backbone.

They introduce a novel model training scheme that proceeds along two distinct flows:
- standard, utilizing coarse visual encoder of LMM on standard textual, spatial, and image inputs; and
- visually-enhanced, comprised of an external visual encoder that accepts "augmented" images where entity's bounding boxes are colored in accordance with entity type.

The resulting encodings are then passed independently through the rest of LMM (if I understand correctly), producing two independent predictions for the span's entity labels, and the final tagging is obtained via Viterbi decoding over predicted probability distributions.
Standard supervised loss against the gold labels is combined with the consistency loss between the two representations.

Given the asymmetry in the inputs to the two learning flows, the authors hope to refine LMM encoder to capture finer visual features, which is evidenced by the method consistently outperforming standard LayoutLM of different versions and other baselines on FUNSD, SROIE and SEABILL datasets.

The visually-enhanced training flow can be detached during inference.

Additionally, the authors conducted several follow up studies, including
- empirical comparisons between different divergence measures and related training schemes: mutual learning and R-drop,
- visualization of learned representations that demonstrates stronger coupling between visual and textual embeddings,
- evaluation of the method in the low-resource and un-pretrained settings, all of which demonstrate consistent improvement over baseline.

**Questions For The Authors:**

Question A: The method acronym is a bit awkward, why not simply name it VACL?

Question B: The color painting step is not sufficiently explained and requires some guesswork on the part of the reader. Referred to by different terms throughout the paper: color priors, synthetic painted images, colorful prompt paints, reverse aligning -- some of which are never mentioned anywhere else in the paper.

Question C: The overall architecture diagram is beautiful but is somewhat confusing. Especially separating the encoder in box 2 from LMMs in box 3, which in my understanding are all part of the same LMM. The detach relations are also hard to follow. It appears as the robust outer encoder is not training? Yet, why color assigning and reverse aligning steps are not detached?

**Reasons To Accept:**

The paper introduces a novel asymmetric consistency training scheme that enhances visual representations learned by multimodal encoders and is empirically shown to outperform a comprehensive set of baseline models.

The authors back the quantitative results with thorough and insightful follow up experiments looking at competing training schemes, visualizing learned embeddings, settings with low data volume (in which the approach especially shines) and without pre-training.

The novel learning scheme can be adapted to an existing LMM backbone.

The paper is generally well written and uses visualizations effectively.

**Reasons To Reject:**

On the conceptual level: The authors claim to bridge the gap between visual feature extraction and entity detection. As evidence that standard visual encoder is limited, authors cite (Cooney et al., 2023) but it merely asserts text has highest predictive power compared to layout geometry but doesn't specifically indicate models lacks capacity for leveraging rich visual features. They additionally argue LayoutLM receives already preprocessed text tokens and are not trained to detect fine grained features such as ones needed for entity detection. Yet, their own addition merely augments images with color patches corresponding to entities, which one could conceivably argue isn't particularly fine-grained either.

It is not clearly articulated why performance improvement would carry over when input images are not augmented with color patches at test time.

**Reproducibility:**

4: Could mostly reproduce the results, but there may be some variation because of sample variance or minor variations in their interpretation of the protocol or method.

**Reviewer Confidence:**

4: Quite sure. I tried to check the important points carefully. It's unlikely, though conceivable, that I missed something that should affect my ratings.

**Typos Grammar Style And Presentation Improvements:**

Typo:
084 et al., 2021b), we propose a novel vision-enhance

Clarity:
097 The inputs are asym-
098 metric in the visual modality, with the original doc-
099 ument images input to the standard flow, while im-
100 ages to the vision-enhanced flow are the synthetic
101 painted images.

It is 150
worth noting that the input images for the vision- 151
enhanced flow are colorful prompt paints,  152

170 where PY
171 tributions output by the standard learning flow and
172 the extra vision-enhanced flow, which are the la-
173 tent outputs after softmax normalization (i.e., soft
174 label).

---

> ### Author Rebuttal · Authors · 2023-08-28
>
> We are grateful for the reviewer’s valuable feedback!
>
> ***Response-Q1: "On the conceptual level."***
>
> Although the method is simple, we have a strong motivation to pursue this approach. Firstly, the idea is inspired by the color bar stimuli used in human visual attention experiments in cognitive science. We wanted to investigate whether similar color cues would impact the attention of multimodal AI-based machine document understanding models, just as the cognitive experiment that prompts participants to focus on a specific area. Of course, the final experimental conclusion revealed that the results deviated somewhat from our initial expectations. The experiment demonstrated that color did not have a particularly significant impact (for instance, merely altering the grayscale could also produce some effect), but rather the incorporation of entity-level label prior information would enable the machine to learn more robustly.
>
>
> ***Response-Q2:explanation for performance improvement without color patches***
>
> This approach to consistency learning we propose is well able to transfer entity-level priors to the standard model. As illustrated in the color assignment process in Figure 2, the allocation of color patches is grounded in truth, signifying that the input image within the vision-enhanced flow encompasses the mapping relationship between the entity and the label that the standard model is required to learn. Fundamentally, as elaborated in Appendix A.4, we acquire this mapping by converging the multimodal output of the standard model with the vision-enhanced flow through the application of consistency learning. Consequently, even in the absence of color patch augmentation in the input images, the standard model can still effectively learn the representative features of entities and relative relationship between entities.
>
>
> ***Response-Q3/A: acronym***
>
> Although VACL is an impressive name and shorter than VANCL, VANCL possesses a straightforward and clear meaning. Therefore, we suggest that VANCL is a more suitable title for this work.
>
> ***Response-Q4/B: Terminology issue***
>
> Regarding the terminology issue, we appreciate your suggestion and will further optimize certain descriptions to enhance the clarity and comprehensibility of the article.
>
> In order to adhere to the 8-page limit, some content might not have been elaborated upon sufficiently. We kindly request the chance of revision to provide a more detailed explanation of the color correspondence and the relationship between terminologies.
>
> ***Response-Q5: the overall architecture diagram is relatively abstract, need some guesswork***
>
> We agree that the overall architecture diagram is somewhat abstract, which may require a certain degree of interpretation. In response to the feedback received during the previous round of ACL review, we have simplified the illustration as suggested. As a result, we have endeavored to depict the core concept, despite the fact that the implementation entails more intricate details when dealing with LayoutLM, LayoutLMv2, and LayoutLMv3, each of which possesses distinct architectures.
>
> The robust outer encoder is indeed involved in the training process.
>
> In the case of LayoutLM, the encoder and the main LMM-transformers are separate components. However, for LayoutLMv2/v3, the LMMs encompass the encoder, which is accurate.
>
> Regarding the detach relations, it is essential to understand that we detach the additional vision-enhanced flow. Consequently, there are no color assignment or reverse alignment steps following the detachment.
>
>
> ***Response-Q6: Typos***
>
> Thank you. We will correct them!

---

### Meta-Review · Area_Chair_Le2n · 2023-09-19

**Recommendation:** 4

**Metareview:**

In this paper, the authors address the challenge of extracting information from documents containing visual and textual elements. They focus specifically on the task of identifying entities in form-like documents that have a rich visual layout. The authors argue that existing multimodal information extraction systems, such as LayoutLM, have limitations in effectively utilizing fine-grained visual features. To overcome this limitation, they propose a new objective named visually asymmetric consistency learning (VANCL), which can be used in conjunction with any multimodal encoder backbone. The experimental results have shown the effectiveness of the proposed method. Although some reviewers have raised some concerns about some statements in the experiment, all of them generally agree with the effectiveness of the proposed method.

---

### Decision · Program_Chairs · 2023-10-07

**Decision:**

Accept-Main

**Comment:**

In this paper, the authors address the challenge of extracting information from documents containing visual and textual elements. They focus specifically on the task of identifying entities in form-like documents that have a rich visual layout. The authors argue that existing multimodal information extraction systems, such as LayoutLM, have limitations in effectively utilizing fine-grained visual features. To overcome this limitation, they propose a new objective named visually asymmetric consistency learning (VANCL), which can be used in conjunction with any multimodal encoder backbone. The experimental results have shown the effectiveness of the proposed method. Although some reviewers have raised some concerns about some statements in the experiment, all of them generally agree with the effectiveness of the proposed method.